# Magnetic Resonance Imaging in Clinical Trials of Diabetic Kidney Disease

**DOI:** 10.3390/jcm12144625

**Published:** 2023-07-11

**Authors:** Iris Friedli, Seema Baid-Agrawal, Robert Unwin, Arvid Morell, Lars Johansson, Paul D. Hockings

**Affiliations:** 1Antaros Medical, BioVenture Hub, 43183 Mölndal, Sweden; 2Transplant Center, Sahlgrenska University Hospital, University of Gothenburg, 41345 Gothenburg, Sweden; 3AstraZeneca R&D BioPharmaceuticals, Translational Science and Experimental Medicine, Early Cardiovascular, Renal & Metabolic Diseases (CVRM), Granta Park, Cambridge CB21 6GH, UK; 4MedTech West, Chalmers University of Technology, 41345 Gothenburg, Sweden

**Keywords:** magnetic resonance imaging, chronic kidney disease, diabetic kidney disease, kidney failure, clinical trials, surrogate endpoints, non-invasive biomarkers, multiparametric magnetic resonance imaging

## Abstract

Chronic kidney disease (CKD) associated with diabetes mellitus (DM) (known as diabetic kidney disease, DKD) is a serious and growing healthcare problem worldwide. In DM patients, DKD is generally diagnosed based on the presence of albuminuria and a reduced glomerular filtration rate. Diagnosis rarely includes an invasive kidney biopsy, although DKD has some characteristic histological features, and kidney fibrosis and nephron loss cause disease progression that eventually ends in kidney failure. Alternative sensitive and reliable non-invasive biomarkers are needed for DKD (and CKD in general) to improve timely diagnosis and aid disease monitoring without the need for a kidney biopsy. Such biomarkers may also serve as endpoints in clinical trials of new treatments. Non-invasive magnetic resonance imaging (MRI), particularly multiparametric MRI, may achieve these goals. In this article, we review emerging data on MRI techniques and their scientific, clinical, and economic value in DKD/CKD for diagnosis, assessment of disease pathogenesis and progression, and as potential biomarkers for clinical trial use that may also increase our understanding of the efficacy and mode(s) of action of potential DKD therapeutic interventions. We also consider how multi-site MRI studies are conducted and the challenges that should be addressed to increase wider application of MRI in DKD.

## 1. Introduction

Chronic kidney disease (CKD) associated with diabetes mellitus (DM) is known as diabetic kidney disease (DKD) and occurs in approximately 30–40% of people with type 2 DM (T2DM) [1,2]. The global DKD burden is expected to rise in line with the increasing prevalence of T2DM [1,2]. This represents a serious and growing healthcare problem since DKD is associated with increased morbidity and mortality and is a leading cause of kidney failure (KF) worldwide that requires kidney replacement therapy (dialysis or transplantation) [1,2].

Early identification of people at high risk of developing DKD and its progression can facilitate timely treatment intervention and prevent KF; however, predicting DKD evolution remains difficult because its progression is so variable, particularly in people with T2DM [2]. Current biomarkers do not provide insights into underlying DKD pathophysiology, the degree of anatomical damage, nor allow risk stratification. There is a need for novel, sensitive, and reliable non-invasive biomarkers that can improve timely diagnosis and prognosis for patients, as well as aid disease monitoring, and can also serve as bio-markers in clinical trials of new treatments for DKD. This may involve the dynamic and rapidly evolving area of MRI.

## 2. Diagnosing DKD

In clinical practice, a diagnosis of DKD as a cause of CKD in DM is based on a reduced calculated or estimated glomerular filtration rate (eGFR) in the presence of albuminuria. eGFR is calculated using a formula based on serum creatinine and/or cystatin C. Albuminuria is generally considered a hallmark of DKD, but it is not exclusive to DM and occurs in many forms of CKD. Despite recent advances, there are no established biomarkers to monitor kidney disease progression except for eGFR and albuminuria, and even these provide only a rough guide to the extent of kidney damage [3]. The measured glomerular filtration rate (mGFR) using plasma clearance of filtration markers is a more accurate measure of kidney function than eGFR but is time-consuming and cumbersome for both screening and routine use in ambulatory care [4].

The spot urine albumin to creatinine concentration ratio (UACR) is a reasonable measure of albumin excretion at a population level. However, at an individual level the UACR remains highly variable and is sensitive to acute haemodynamic changes that affect the GFR [5], as seen typically when starting antagonists of the renin–angiotensin–aldosterone system (angiotensin-converting-enzyme inhibitors (ACEis), angiotensin receptor blockers (ARBs), or mineralocorticoid receptor antagonists (MRAs)), as well as sodium-glucose cotransporter 2 inhibitors (SGLT2i), all of which can reduce UACR and cause an early reduction in GFR.

Kidney biopsy is the current “gold standard” for diagnosing kidney disease. The extent of tubulointerstitial fibrosis (and not glomerular pathology) seen in a renal biopsy specimen from a patient with DKD is the best predictor of loss of kidney function, disease progression, and likely outcome, whereas GFR per se does not accurately reflect the degree of fibrosis [6,7]. The procedure, however, is invasive and carries a risk for patients of both bleeding, especially in those with late-stage (4/5) DKD and small kidneys, and, in very rare cases, even loss of a kidney. Kidney biopsy is also associated with a risk of bias because of the highly selected patients in whom it can be clinically justified and performed, as well as with sampling error because the kidney is not a homogeneous structure. In fact, DKD patients rarely undergo a kidney biopsy for diagnosis or monitoring unless there are unusual clinical features, such as heavy proteinuria or a sudden and unexpected decline in eGFR [8]. Due to these limitations, kidney biopsy is not suitable for long-term serial monitoring of disease progression or a response to therapy [9]. Thus, DKD patients are still mostly staged prognostically on the basis of both eGFR and the presence and level of albuminuria (from G1, A1, normal eGFR, no albuminuria to G5, A3, eGFR < 15 mL/min/1.73 m^2^ and albuminuria > 300 mg/gCr) [10]. Importantly, significant kidney fibrosis can occur without a detectable change in GFR [6,11] because the kidney can partially compensate for any loss of function (in part through nephron hyperfiltration), which can complicate diagnosis.

Other clinical methods of assessing kidney disease include ultrasound imaging, conventional computer tomography (CT), and, occasionally, magnetic resonance imaging (MRI) [12]. In clinical practice, however, these standard imaging methods tend to be used to exclude other diagnoses (e.g., renal carcinoma or atherosclerosis) rather than evaluate disease progression per se and currently are of limited value prognostically or in patient stratification.

An updated definition and classification of CKD that includes DKD was set out in the Kidney Disease Improving Global Outcomes (KDIGO) guidelines in 2012 [10] with the stages of CKD defined according to the GFR and albuminuria categories [13]. 

## 3. DKD Phenotypes and Stages of Kidney Disease

The original description of DKD highlighted that renal size is markedly increased with a concomitant increase in glomerular filtration (GFR) in both humans and experimental models early in T2DM [14]. These changes may stem from metabolic effects that are only slowly reversible since insulin treatment can lead to a reduction in both renal size and GFR over 3 months [14].

Various DKD phenotypes are well recognized, and other pathways to KF independent of albuminuria have been proposed, indicating different pathophysiologies not readily detected by conventional biomarkers [15,16]. Beyond the classic albuminuric presentation, a non-proteinuric phenotype of DKD occurs in 20–40% of T2DM patients; this phenotype progresses more slowly than that in patients with albuminuria yet still carries a greater risk of death and major cardiovascular outcomes than in patients without DKD [17]. Typically, non-albuminuric DKD is characterized by tubulointerstitial injury and fibrosis, which suggests that the histological injury occurs in the vascular and interstitial compartments rather than in the glomeruli seen in albuminuric DKD [18].

Importantly, people with T2DM may also have other causes of CKD with or without the typical histological features of DKD. In a broader sense, all the components of the metabolic syndrome (impaired fasting glucose, elevated blood pressure, obesity, increased triglycerides, and reduced HDL-C) are known to be individually associated with CKD in varying degrees with elevated blood pressure showing the strongest association [19]. Mixed comorbidities occur and since the diagnostic strategy is one of exclusion, patients may be diagnosed as having DKD, but have comorbidities or a disease unrelated to DKD that also affect kidney function. Metabolic dysfunction and obesity drive multi-organ dysfunction, including non-alcoholic fatty liver and myocardial diastolic dysfunction, which ultimately lead to end-organ dysfunction, such as CKD [20,21,22]. The cause of progression to end-organ failure and KF in CKD is multifactorial; pathological backgrounds are likely to overlap yet can also be uniquely associated with different rates of progression (Figure 1). Moreover, dysfunction of one organ, such as the kidney, can negatively impact other organs and lead to further organ failure, for example, the heart [23].

Episodes of acute kidney injury (AKI) in patients with DM and DKD are also thought to contribute to disease progression and eGFR decline [24]. Clinical AKI is defined by KDIGO as an increase in serum creatinine of >0.3 mg/dL (>26 μmol/L) within 48 h or an increase to more than 1.5 times the baseline serum creatinine within 7 days [25]. Many, but not all, patients eventually recover their baseline kidney function over days or weeks; however, even a seemingly recovered episode of AKI carries an increased risk of developing CKD later, the so-called “AKI to CKD transition”, especially in patients already known to have some impairment of kidney function with or without pre-existing albuminuria [24,26,27]. 

A major disconnect between glomerular function and renal structure is well described, such that greater than 50% loss of functional mass is required for an increase in serum creatinine [6]. Given that the prevalence of overt CKD in most populations is at least 10% [8,28], there is likely to be an even greater prevalence of subclinical CKD, defined as severe parenchymal damage in the presence of a normal serum creatinine level. Thus, a return to previous or normal creatinine-baseline values after AKI provides no insight into how much acute parenchymal damage has been sustained [6].

## 4. Endpoints for Clinical Trials in DKD

Compared with other clinical specialties, such as cardiology, until recently, relatively few large-scale clinical trials have been conducted in nephrology [29], and these are usually in a small number of selected patients and often those on kidney replacement therapy (i.e., dialysis or transplantation). One reason is that drug approval and registration regulatory bodies require clinically meaningful clinical trial endpoints in Phase 3 to demonstrate the efficacy of a treatment and reflect how a patient feels, functions, and survives. Endpoints can be objective measures, for example, a clinical sign, and/or subjective, such as quality of life measures [30].

Following a landmark publication [31] as part of a joint working group with the National Kidney Foundation (NSF) and FDA in the US and the European Medicines Agency (EMA), a proposal was made to use a 30% reduction in albuminuria from baseline at 6 months and/or a reduction in eGFR slope decline of 0.5–1 mL/min/1.73 m^2^ over 2 years as surrogate endpoints for efficacy in intervention trials in DKD/CKD. This was based on an extensive series of meta-analyses of previous observational and treatment intervention studies in DKD/CKD [32,33]. While both the FDA (more so) and EMA have been broadly supportive of these new measures as surrogates [34,35], neither endpoint has been accepted yet as an approved endpoint for final drug registration in lieu of the “hard” renal endpoints of death, dialysis, or transplantation. 

Use of albuminuria or proteinuria reduction is looked on more favourably as an endpoint in smaller and shorter Phase 2 trials in patients who have significant albuminuria or proteinuria [36,37]. Phase 2 clinical trials are conducted in around one hundred patients with the target disease to define the most efficacious dose, delivery route, and frequency of administration, as well as safety. Early reductions in albuminuria (or proteinuria) are being widely used to test new molecular entities in DKD, even for as short a time as 3 months of treatment duration. However, selecting albuminuria as the endpoint excludes non-proteinuric DKD patients from Phase 2 clinical trials. These patients progress more slowly than those with albuminuria/proteinuria [15] yet comprise up to 40% of those reaching KF and often have never undergone a kidney biopsy [38]. As, typically, only high-risk patients or “fast progressors” are selected for early clinical trials, a significant proportion the DKD patient population is excluded until Phase 3, which can introduce bias and account for poorer efficacy seen in Phase 3 clinical trials [10].

Choosing endpoints that are well suited to short-term and exploratory Phase 2 clinical trials in relatively small numbers of representative patients is crucial. Ideally, these chosen endpoints would also have an evidence base linking the biomarker to a relevant clinical outcome of interest. This is the major challenge confronting the early clinical development of all drugs and particularly in finding effective treatments for DKD or CKD, neither of which are strictly single diseases. This also explains the current focus on identifying easily measurable “biomarkers” that can be used to stratify patients based on an underlying disease mechanism (diagnostic), risk of disease progression (prognostic), likely response to a given treatment (predictive), and treatment efficacy to enrich a trial with the optimal patient population. However, such novel biomarkers are still exploratory; apart from using albuminuria and the eGFR slope for efficacy, the soluble biomarkers currently approved for use in clinical trials are for safety monitoring only.

New drugs, including the SGLT2i and MRAs, have revolutionized DKD treatment and help to preserve kidney function [39]. Better biomarkers, however, are needed to predict or to confirm a response to a given treatment and to identify both novel treatment targets and mechanisms of action of therapeutic interventions [40]. While progress has been made in identifying some novel soluble biomarkers in blood or urine, there are still no suitable agreed biomarkers [41].

## 5. A Potential Role for and Opportunities with MRI

Interest has turned recently to the use and potential of MRI and the detailed structural and functional readouts it can provide to non-invasively assess and quantify pathophysiological changes in CKD [42,43,44] (Figure 2). As one of the foremost imaging techniques to aid medical diagnoses, MRI is the method of choice for diseased (and normal) soft tissue because the contrast can be “tailored” using multiple “weighting” or “sensitization” techniques. Thus, MRI can distinguish between tissue types and organs despite their very similar water content. The contrast generated by these sensitization techniques reflects aspects of the physicochemical environment of the water molecules in the tissue. Tissue properties, such as tissue microstructure, composition, metabolism, function, and gross morphology, can be assessed with quantitative imaging biomarkers. However, given that DKD is usually diagnosed using a simple blood test of kidney function and, in some cases, a spot urine albumin test, it seems unlikely that MRI will be used soon to diagnose DKD in normal clinical practice. Rather, MRI and other imaging techniques will continue to be used to exclude an alternative diagnosis in questionable cases.

The real scientific advantages of MRI are that unique aspects of the pathophysiology can be quantified; the lack of exposure to ionizing radiation or radioactivity means subjects (normal volunteers or patients) can be scanned repeatedly; in clinical trials, pre-treatment scans allow each subject to act as their own control and can be used to screen subjects for study inclusion. Importantly for the DKD population, renal MRI may complement or even provide an alternative to kidney biopsies with the advantage of separate evaluation of both kidneys in their entirety, thereby avoiding biopsy-associated sampling bias and permitting detection of regional variations. The high-spatial detail allows both the cortex and the medulla to be visualized. These advantages are seen in clinical trials, where MRI can help to elucidate the mechanism of action of new drugs.

One recent advance in the MRI field is the multiparametric MRI (mpMRI) biomarker, defined as two or more imaging biomarkers that can be used collectively or combined to diagnose, give a prognosis, or monitor a disease. Importantly, mpMRI may provide more comprehensive information on the macrostructure (kidney parenchyma and cortex volume), haemodynamics of renal blood flow (RBF) and perfusion [45], oxygenation, and microstructure (including kidney fibrosis and inflammation) than individual imaging endpoints [46] (see Figure 2). mpMRI provides a variety of imaging contrasts that can differentiate pathological from healthy tissues according to biophysical changes. Despite known limitations in terms of biological specificity, it seems plausible that a direct observation of parenchymal changes in situ can detect disease progression well before it manifests itself as changes in blood or urine. As a change in fibrosis is the best measure and predictor of CKD progression, an ability to detect this non-invasively would be a major advance and more closely related to the treatment target [47].

**Figure 2 jcm-12-04625-f002:**
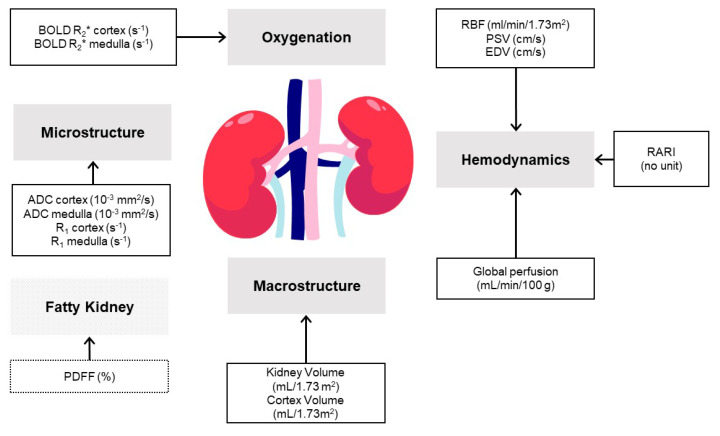
Recommended Kidney Magnetic Resonance Imaging Endpoints. MRI biomarkers (units of measurement) associated with kidney macrostructure, haemodynamics, oxygenation, and microstructure. A multiparametric approach comprises a composite endpoint based on several of these parameters. Based on Makvandi et al. [48]. ADC = apparent diffusion coefficient; ASL = arterial spin labelling; BOLD-R_2_* = Blood Oxygenation Level-Dependent apparent relaxation rate; EDV = end diastolic velocity; MRI = magnetic resonance imaging; PSV = peak systolic velocity; R_1_ = apparent relaxation rate; RARI = Renal Artery Resistive Index; RBF = renal blood flow.

The converging interests of MRI physicists, radiologists, nephrologists, drug developers, transplant surgeons, physiologists, and pathologists have given rise to a dynamic and multidisciplinary community of researchers with a common interest in renal MRI biomarkers. The international, pan-European, multidisciplinary research network PARENCHIMA [49] (European Cooperation in Science and Technology [COST] Action) was founded in 2017 with the aim of eliminating key barriers to the wider evaluation, commercial exploitation, and clinical use of renal MRI biomarkers. Its work now continues as renalmri.org (renalmri.org, accessed on 10 July 2023) and provides guidelines and updates on MRI biomarker use in CKD. Following the fourth international meeting on renal MRI that attracted scientists from across these disciplines (ISMRM 2021 Workshop on Kidney MRI Biomarkers: The Route to Clinical Adoption), the broadening interest in MRI led to the publication of several systematic reviews and consensus-based technical recommendations [45,50,51,52,53,54,55,56,57,58,59,60,61]. MRI biomarkers are also increasingly visible in the nephrology literature [43].

## 6. Employing MRI Endpoints in Clinical Trials

The FDA’s Guidance for Industry Standards for Clinical Trial Imaging Endpoints [62] represents its current thinking on the use of imaging endpoints in clinical trials intended to support drug and biological product approval. Imaging acquisition and interpretation standards can be divided into a medical practice standard and a clinical trial standard. Briefly, the medical practice imaging standard may rely on an investigator’s response to a clinical question determined by any available medical practice method, for example, “What is a patient’s cardiac ejection fraction?” The clinical trial imaging standard specifies the standardization of image acquisition and analysis to enhance the ability to detect a drug effect and to verify data integrity. Clinical trial imaging standards exceed those used in medical practice. For diagnosis, imaging is often used to inform dichotomous diagnostic decisions (for example, the presence of cancer = yes/no), whereas, in drug development, the purpose is usually to monitor a change over time of a continuous variable that increases the demand on the imaging biomarker in terms of precision and power.

The best-known example of a prognostic MRI biomarker is total kidney volume (TKV) in autosomal dominant polycystic kidney disease. TKV is the only MRI biomarker to have been approved by the FDA as a prognostic enrichment biomarker to select patients for interventional clinical trials who are at higher risk for a progressive decline in renal function [63] and is most accurately measured by MRI [64]. Apart from TKV, imaging endpoints are not currently used as regulatory endpoints but may be used in Phase 2 studies or mechanism of action studies conducted in Phase 3, where there is evidence to link the biomarker with underlying biology or with clinical endpoints. There is a clear need for new, qualified biomarkers for clinical trials that might be MRI in nature. As MRI currently is of less value in Phase 1 studies that comprise the “first time in man” administration of a drug to healthy human volunteers (usually males) principally to assess safety by gradual dose escalation and evaluation of its absorption, metabolism, accumulation, and duration of action rather than efficacy, we will focus our discussion on Phase 2 and 3 clinical studies. 

Significant questions regarding MRI should be considered: will it provide unique, decision-influencing data unobtainable by other means and/or offer higher quality information, whether unique or not, and thus increase the precision of measurements producing robust data using fewer subjects or shorter study durations. Moreover, cost, convenience, time taken, and opportunity for re-analysis are important considerations in clinical trials and the advantages of expensive, resource-intensive, and patient-demanding techniques need to be clear and compelling. 

A common hindrance to the inclusion of imaging procedures in clinical trials is cost. In 2015–2016, the median per patient cost in pivotal clinical trials for novel therapeutic agents approved by the FDA was USD 41,117 and USD 3562 per patient visit [65]. Introducing novel endpoints with improved repeatability into clinical trials can reduce patient numbers and/or trial duration, thereby reducing their overall cost. Thus, the additional cost of including imaging endpoints must be seen in relationship to the “per patient” costs and the potential for reducing patient numbers or visits. The ability to accelerate and render a clinical trial safer by reducing drug exposure in patients is a highly attractive opportunity for drug development, and offering a medical read of the scan can facilitate recruitment to a study. Ultimately, however, medical imaging needs to enhance the ability to quantify the impact of drugs on human health compared to conventional or invasive clinical assessments.

To date, approximately 40 publications have reported the effects of various interventions relevant to understanding the pathophysiology of DKD that have utilized renal MRI endpoints. Interventions include water loading, loop diuretics, thiazide diuretics, non-steroidal anti-inflammatory drugs, calcineurin inhibitors, ACEis, ARBS, direct renin inhibitors, calcium channel blockers, norepinephrine, high- or low-salt diets, glucose, glucagon-like peptide-1 (GLP-1) agonists, and SGLT-2i. Some of these studies have been conducted in healthy volunteers, whereas others have been conducted in patients with DKD, CKD, hypertension, renal artery stenosis, DM, or heart failure. We have summarized a small selection of examples in Table 1. These studies show that changes in renal MRI biomarkers after intervention can be seen with relatively few healthy volunteers or patients and that renal MRI can measure endpoints that reflect changes in the underlying pathophysiology after intervention.

## 7. MRI Methods

It is important to know which MRI parameters can detect and diagnose early kidney injury; point reliably to a predominant underlying pathological process; predict a decline in kidney function and the likely patient outcome; track disease progression; and monitor treatment response. We will attempt to address these questions by reviewing the MRI techniques currently available and their potential as renal MRI endpoints in intervention studies (Figure 2; Table 1).

### 7.1. Kidney Macrostructure

Ultrasonography is often used in clinical practice to evaluate patients with CKD to rule out potentially reversible causes; to estimate kidney size to decide whether to conduct a renal biopsy; and to obtain prognostic measures [77]. Volume measurements made with MRI are generally considered to be more accurate and precise than ultrasound measures. In people with atherosclerotic renovascular disease, kidney parenchymal volume (KPV) measured with 3D MRI was better correlated to single-kidney GFR (r = 0.86, *p* < 0.001) than renal bipolar length and parenchymal and cortical thicknesses [78]. Indeed, KPV adjusted for Body Surface Area correlated with mGFR (r = 0.61, *p* < 0.001) in a study of DKD patients and healthy volunteers [48].

In both humans and animal models, markedly increased renal size and function appear to occur early in T2DM [14]. Normoglycemic individuals have been reported to have a TKV of 280 mL, pre-diabetic patients 304 mL, and T2DM patients 321 mL [79]. Moreover, in a large-scale cross-sectional study of 37,450 UK Biobank participants, T2DM patients had a larger KPV than non-T2DM patients but showed a faster KPV decline independent of lean tissue volume differences. The faster decline in T2DM may be explained by increased hyperfiltration and oxidative stress that occurs in the kidneys of people with T2DM [80]. Liraglutide treatment is reported to significantly decrease KPV compared to placebo in T2DM patients (see Table 1) [75].

In contrast to the renal hypertrophy seen in early T2DM, significant reductions in KPV for DKD patients with CKD stages G3–4 (176 mL/1.73 m^2^) compared to age- and gender-matched healthy volunteers (218 mL/1.73 m^2^) have been reported [48]. Repeatability for measurements taken 2 weeks apart had a coefficient of variation (CV) of 7% and an intra-class correlation coefficient (ICC) of 0.89 [48]. A study performed in non-DKD, CKD patients with stages G3–4 showed a similar TKV of 170 mL/1.73 m^2^ with a 4% CV of the measurements [81]. Interestingly, AKI patients have shown an increased TKV at the time of AKI; however, TKV returned to the normal range, or even decreased to the range for CKD patients, in most AKI patients after 1 year [82]. Episodes of AKI can contribute to DKD progression and TKV may be used to monitor these excursions. 

Manual segmentation of kidneys on MR images is tedious and operator dependent [60], and a recent review of renal image segmentation techniques has highlighted limitations that might hinder clinical translation [60]. Image segmentation is an important step for TKV assessment but can also be used to derive kidney parenchyma contours, as well as cortex and medulla volumes while excluding renal tumours or cysts. These contours can be used subsequently in the analysis of functional MRI techniques, including perfusion, diffusion, or BOLD. As DKD progresses, the corticomedullary contrast seen on MRI decreases [83], making it difficult to derive separate cortex and medulla volumes in people with DKD. This remains a challenge. Contrast-enhanced MRI can be used to produce good contrast between the cortex and medulla; however, the risk of nephrogenic systemic fibrosis caused by contrast agent use should not be neglected. The corticomedullary contrast ratio has been improved significantly on inversion recovery Steady-State Free Procession (SSFP) MRI without contrast compared to conventional, in-phase, T_1_-weighted, gradient-echo MRI, and corticomedullary contrast ratio was positively correlated with eGFR [84]. Future studies, therefore, may allow us to study the effects of DKD progression or therapeutic interventions on cortical volume.

### 7.2. Kidney Haemodynamics

The way in which DKD affects kidney haemodynamics can be complex. The kidney can compensate for structural changes (e.g., reduction in total nephron number, interstitial fibrosis, and/or vascular rarefaction) that may precede any GFR changes in early-stage CKD patients by increasing RBF and hyperfiltration at the glomerulus [85,86]. Structural changes that occur later as DKD develops may subsequently affect RBF, likely reducing it due to the increased resistance of renal microcirculation. The kidney is very effective in maintaining glomerular pressure and filtration rate in healthy people, although such autoregulation may be gradually lost as CKD progresses [45]. Moreover, CKD patients receive a wide range of drugs that alter renal function and RBF. RBF has long been known to be influenced by protein-rich meals and hydration state [87] and can be increased by up to 50% post-prandially such that diet and hydration state must be controlled to ensure comparability and repeatability of renal haemodynamic measurements. 

Measurement of RBF by infusion of para-aminohippurate (PAH) and blood sampling over several hours has long been the “gold standard” for determining kidney haemodynamics. The renal extraction rate of PAH is assumed to be 85% in healthy people, although this decreases in those with renal impairment, with the individual reductions being rather unpredictable [88]. PAH infusion is also burdensome for patients and cannot differentiate single kidney blood flow. Colour Doppler ultrasonography, an easily accessible, alternative technique that can differentiate single kidney blood flow, is user dependent and presents technical challenges in accurate evaluation of flow measurements in overweight patients [45]. MRI may be a superior alternative. The main MRI methods used to evaluate kidney haemodynamics are phase-contrast MRI (PC-MRI), arterial spin labelling (ASL), and dynamic-contrast-enhanced MRI (DCE-MRI) (see Table 1). A recent study compared all three techniques in T2DM patients and concluded that the repeatability of PC-MRI measurements supported its use as a reference method for MRI of RBF [89]. Furthermore, the comparison showed that while DCE-MRI and ASL measurements are unbiased, they showed poor precision relative to PC-MRI [89].

#### 7.2.1. Phase-Contrast-MRI

PC-MRI is a non-contrast-enhanced MRI technique that allows blood velocity and flow to be determined in a specific vessel during the cardiac cycle within a few minutes. Notably, PC-MRI directly measures RBF, unlike alternative MRI techniques such as ASL, where total renal perfusion depends on labelling efficiency and the T_1_ of blood and tissue that can introduce bias into the perfusion measurements. The first step in PC-MRI is an angiogram acquisition to enable planning of a PC-MRI acquisition perpendicular to the renal arteries and prior to any bifurcation. Phase and magnitude images are then acquired for each renal artery, and RBF (mL/min) is computed by multiplying the renal artery area (more accurate than cross-section) and mean blood velocity. While acquisition of the 2D PC-MRI datasets takes one breath-hold for each kidney, planning can be challenging and time-consuming. Total RBF per kidney should be reported in millilitres per minute for the sum of the left and right kidneys, although RBF can be reported for individual kidneys if, for example, one kidney is stenotic. In the case of multiple renal arteries, RBF through the main and accessory arteries should be combined [52].

Typically, RBF is 1.1 L/min [90]. PC-MRI has shown that RBF is significantly decreased in both mixed CKD patient groups [46,81,91] and in a DKD group [48] compared to healthy volunteers. Moreover, RBF has been shown to differentiate between stages G3 and G4/5 of DKD, with an AUC of 0.88, and *p* = 0.004 [48]. RBF measurements using PC-MRI correlate well with “gold standard” methods, such as PAH infusion [45]. When examined 1–2 weeks apart, reproducibility of respiratory-gated PC-MRI was generally good in CKD patients and healthy volunteers with CVs of 12.9% and 8.3%, respectively [92], although one study reported an intra-subject CV of 18% for CKD patients [81]. In DKD patients and healthy volunteers examined 2 weeks apart, RBF had a CV of 7% and an ICC of 0.97 [48]. A CV of 6% was seen in healthy volunteers despite the relatively long interval to complete four repeat scans (4 months on average) [89].

Global kidney perfusion (mL/100 g per min) can be obtained by dividing RBF by kidney volume and multiplying the result by 100 [93]. Highly significant decreases in global perfusion have been demonstrated both in DKD patients versus healthy volunteers [48] and in a non-diabetic CKD population [81]. The lower global perfusion values reported by Buchanan et al. [81] compared to Makvandi et al. [48] may be associated with methodological differences, such as using TKV [81] rather than KPV [48] in global perfusion calculations.

PC-MRI measures blood velocity throughout the cardiac cycle and, therefore, haemodynamic biomarkers, such as end diastolic velocity (EDV), peak systolic velocity (PSV), and the Renal Arterial Resistive Index (RARI) (see Figure 2), are also available. Notably, EDV and RARI enabled healthy volunteers to be differentiated from DKD patients, and both EDV and PSV could distinguish DKD patients with stage G3 versus stage G4/5 [48]. These are novel MRI-biomarkers that may provide additional insights into DKD pathophysiology. A number of derivative biomarkers, such as filtration fraction (FF) and PC-MRI-based renal plasma flow (PC-RPF) [94], can also provide important insights into kidney pathophysiology.

Ultimately, the goal is not to replace readily available GFR measurement with an expensive imaging technique but to provide additional mechanistic data that may be useful to understand the pathophysiology of the underlying disease, as well as the efficacy and/or mechanisms of action of novel drugs fundamental to drug development. A systematic review and a Consensus-Based Technical Recommendation on renal PC-MRI have been coordinated by PARENCHIMA (renalmri.org accessed on 10 July 2023) [45,52], and we have summarized its additional use in specific interventional studies in Table 1.

#### 7.2.2. Arterial Spin Labelling

ASL is an MRI technique that produces a quantitative map of perfusion in the target organ of interest by using arterial blood water as an endogenous contrast agent. Its use in perfusion imaging outside of the brain was reviewed recently [56,95]. The PARENCHIMA consensus paper on kidney ASL recommended that only cortical perfusion should be reported because medullary ASL measurements are not reliable [55]. The advantage of ASL over PC-MRI is that the perfusion image can show heterogeneity in the kidney, which may indicate a lesion. ASL creates perfusion maps by subtracting labelled blood images from control images, and therefore is susceptible to physiological motion, such as breathing and peristalsis. Another challenge is the relatively long acquisition time compared to PC-MRI. 

ASL cortical perfusion was reported to be lower in DKD patients [48] compared to healthy volunteers. Of note, changes in ASL cortical perfusion with no significant decrease in RBF have been reported after infusions designed to expand blood volume, where the decrease in ASL cortical perfusion measured in mL/100 g per min was due to an increase in kidney volume [96]. It has also been speculated that an increase in ASL cortical perfusion with no change in RBF after treatment with a GLP-1 agonist may be due to changes in kidney volume [97]. The intra-subject CV has been reported to be 9% and 31% in healthy volunteers [46,89], 23% in CKD patients [81], and 33% in DKD patients and matched healthy controls [48].

The first renal ASL study was performed in 1995 [98]. To the best of our knowledge, in the intervening years only single-centre, renal ASL studies have been performed. This may be because the MRI vendors offer different ASL techniques, and many imaging centres have developed their own ASL solutions under research agreements, which makes coordinating studies with similar techniques challenging. Whilst ASL is an exciting technique that offers unique insights into kidney perfusion, it may not yet be ready for use in multicentre clinical trials investigating novel drugs, although it has been used in some interventional studies as a renal MRI endpoint (Table 1).

#### 7.2.3. Dynamic Contrast Enhanced-MRI

Gadolinium-based contrast agents are widely used for enhancing the contrast of magnetic resonance (MR) images and improving MRI diagnostic capabilities. Magnetic resonance renography (or renal dynamic contrast-enhanced MR, DCE-MR) depends on the transit of intravenous gadolinium chelates through the parenchyma and collecting system of the kidney. Compartmental analysis of renal tissue enhancement as a function of time has shown promising results for assessing single-kidney function in healthy kidneys and various renal impairments. Renal DCE-MRI endpoints are cortical perfusion (mL/min/100 mL), filtration fraction (%), tubular volume fraction (%), and blood volume fraction (%) [99,100,101].

However, gadolinium-injection-based MRI methods are not commonly used in multicentre clinical trials to assess DKD patients because concerns about the contrast agent translate into use restrictions in this population. Despite advances in the chelate structures that hold the gadolinium, a black box warning still exists for its administration due to potential brain deposition and toxicity in people with kidney disease [102,103], cautioning its use in DKD patients with CKD stages 3–5. Additionally, the high variability and low concordance rates of DCE-MRI measurements [104] hinder the repeatability, accuracy, and precision required for their use to monitor treatment in DKD patients [89]. Moreover, there are no reference-standard methods for translating renal DCE-MRI into clinical trials. 

A comprehensive introduction to renal DCE-MRI methods, as well as acquisition-based motion correction and registration techniques, have been published recently based on work conducted by PARENCHIMA [61,101]. These illustrate current limitations and possible sources of discrepancies, including, but not limited to, differences in image acquisition [105,106,107], motion correction applied [61,108], and kidney segmentation [109,110,111] using DCE-MRI. Moreover, GFR estimates from DCE measurements depend on the kinetic model and data type used [112]. In hypertensive patients, a 6% overestimation to 50% underestimation of single-kidney DCE–eGFR values using the same data across different models compared to standard radionuclide clearance and gamma-camera renography as the reference have been reported [113]. Similarly, DCE–eGFR has been overestimated in hydronephrosis [114] and allograft patients [115], whereas DCE–MRI has underestimated eGFR in chronic liver disease [116] and hypertensive patients [117].

ASL and DCE have been compared for the measurement of RBF in healthy volunteers [118] and T2DM [89] with no significant mean differences reported. However, a recent comparison of RBF measurements evaluated by DCE-MRI, ASL, and PC-MRI in T2DM showed a poor agreement on individual level between these MRI methods [89]. Intra-subject CVs of 15–22% were reported for DCE-MRI-based RBF, tubular flow, and eGFR with a 10-day interval between each measurement [119]. Boer et al. [120] reported a 17% CV for DCE perfusion in healthy subjects using a radial acquisition strategy and a temporal resolution of only 4.1 s.

Thus, the major limitations of DCE-MR, specifically the black box warning associated with the injection of an additional substance (i.e., gadolinium) into DKD patients with CKD stages 3–5 together with the lack of a reference-standard methodology, limit translation of renal DCE-MRI into DKD clinical trials. This leaves an opportunity for future studies to address the variability associated with this technique.

### 7.3. Oxygenation (BOLD)

The kidney BOLD MRI technique uses the paramagnetic properties of deoxyhaemoglobin as an endogenous contrast agent to non-invasively assess deoxyhaemoglobin concentration in tissue [50]. Changes in deoxyhaemoglobin tissue concentration contribute to creating microscopic magnetic field inhomogeneities that can be captured with the BOLD-related apparent relaxation rate R_2_* [s^−1^]. A higher level of deoxyhaemoglobin in blood increases R_2_*. However, the fractional blood volume in tissue and blood hematocrit will also affect deoxyhaemoglobin tissue concentration and thus R_2_*. It was reported recently that kidney cortex and medulla fractional blood volume is significantly decreased in CKD patients compared to healthy controls; cortical R_2_* was the same in both groups. Taking the fractional blood volume into account, it could be shown that the kidney cortex is normoxemic in healthy controls and hypoxemic in CKD [121]. Fractional blood volume was measured using an intravenous iron oxide nanoparticle formulation for treating anaemia in patients with CKD that is not available as a MR contrast agent in standard clinical practice. Fractional blood volume is an important parameter to consider when interpreting BOLD MRI data but has not been available in most clinical trials.

Renal oxygenation is based on a balance between oxygen supply and consumption [122]. Most of the oxygen consumption in the kidney is due to reabsorption of filtered sodium, so, when renal blood flow (and thus renal oxygen delivery) increases, renal oxygen consumption also increases, as there is more filtered sodium to be reabsorbed [123]. If the filtration fraction does not vary significantly, then a near-constant tissue oxygen tension should prevail in the tissue. In renal physiology, the filtration fraction is the ratio of GFR over the renal plasma flow. This suggests that filtration fraction might be one of the main determinants of renal oxygenation status [94]. In healthy volunteers under continuous steady-state infusion of angiotensin II, R_2_* values appeared to be more associated with kidney filtration fraction than GFR, and the measured reduction in RBF was only accompanied by a minor change in cortical R_2_* [94]. The amount of filtered sodium is the product of the GFR and the plasma sodium concentration, and, therefore, plasma sodium concentration may also affect renal BOLD.

A major advantage of using BOLD to indirectly assess renal tissue oxygenation lies in its non-invasive nature compared to direct oxygen pressure measurement using microelectrodes as the only existing alternative and not acceptable for use in volunteers or patients. BOLD has been used to explore the potential renoprotective role of drug interventions, such as GLP-1 agonists [97], lipo-prostaglandin E1 (Lipo-PGE1) [124], or SGLT2i [125,126] (see also Table 1). A single 50 mg dose of dapagliflozin, a medication that inhibits the uptake of sodium and glucose from the filtrate, decreased cortical R_2_* with no change in renal perfusion and RBF, potentially due to reduced workload because of the reduced sodium re-uptake [73] (see Table 1). Consistent with this finding, BOLD was able to capture an improved renal hypoxia in newly diagnosed T2DM patients treated with canagliflozin compared to glimepiride-treated controls [126].

BOLD data should be interpreted carefully because factors other than the fraction of haemoglobin that is deoxygenated may affect the BOLD signal [127,128,129]. Apparent discrepancies in BOLD findings from the same disease state have been reported [81,130,131,132]. For example, a significant increase in the medullary R_2_* was measured after 32 weeks of treatment with empagliflozin alone and in combination with semaglutide, suggesting that renal oxygenation was not improved [76]; notably, however, the haematocrit increased significantly in these groups, so the change in R_2_* may reflect haematocrit instead of oxygenation status (see also Table 1). 

Importantly, R_2_* from BOLD has shown potential as a prognostic biomarker for predicting kidney outcomes and progressive renal function decline in CKD patients after 1–4 years of follow-up [130,133,134]. In the future, BOLD R_2_* could be used to select at risk patients for inclusion in clinical trials or for more intensive monitoring in clinical practice. Indeed, an important milestone for BOLD-MRI application in clinical trials has been the standardization of patient preparation, data acquisition, and analysis protocols to deliver comparable data at most imaging centres and ongoing multicentre initiatives to standardize BOLD-MRI methodology have been supported by PARENCHIMA [50,58]. The key to using BOLD MRI in drug development is the good repeatability of BOLD-MRI measures, as supported by recent studies that reported less than 7% variations for R_2_* test–retest repeatability analyses conducted on different days [46,48,120].

### 7.4. Kidney Microstructure

#### 7.4.1. The T_1_ Relaxation Time

Native T_1_ mapping is a non-contrast-enhanced quantitative MRI technique derived directly from MR relaxometry, in which the tissue contrast is determined by the longitudinal (spin-lattice) relaxation time. Each pixel of the T_1_ parametric map represents how quickly the nuclear spin magnetization returns to its equilibrium state after an RF pulse in the MRI system [135]. Different acquisition schemes, post-processing, and data analysis can be used to deliver T_1_ data, and PARENCHIMA recently published a systematic review and consensus-based recommendations on how to generate T_1_ values [53,59]. 

The T_1_ relaxation time, also known as the relaxation rate R_1_ (R_1_ = 1/T_1_), is a tissue-specific time variable. T_1_ values depend on the environment of water molecules within the tissue, and changes in T_1_ may be useful biomarkers to assess water content abnormalities. Decreased cortical T_1_ has been measured in compensated and decompensated cirrhosis patients [136], whereas increased T_1_ has been measured in IgA nephropathy (IgAN) [137] and higher CKD stages [46,48,81,138]. Furthermore, in CKD patients—and like the diffusion-weighted imaging (DWI)-generated Apparent Diffusion Coefficient (ADC)—T_1_ could potentially separate relatively “low-level” from “high-level” fibrosis [81]. Thus, T_1_ is also very promising for fibrosis assessment and could be an important prognostic biomarker. Native T_1_ values, however, lack specificity and can be influenced by several different factors [138], and, while T_1_ values correlate with interstitial fibrosis [139,140], this also applies to other lesions, including tubular atrophy, chronic vasculopathy, and transplant glomerulopathy [141]. 

As MRI biomarkers of the kidney microstructure, T_1_ and ADC improve fibrosis detection with good cross-validated diagnostic performance when used in combination [139,140] and may, therefore, provide additional information on disease progression in DKD. T_1_, alone or in combination with other biomarkers, performed well in predicting kidney outcome at 18 months [139] and renal function decline at a mean of 2 years [142]. T_1_ was a significant predictor of UACR even when combined with mGFR in a bivariate model [48]. Furthermore, in a renal mpMRI study that included follow-up data on seven AKI patients, a significantly increased T_1_ was measured in AKI patients compared to healthy volunteers despite a small sample size [82]. Interestingly, during recovery from the acute AKI phase, T_1_ and serum creatinine measurements improved and returned to normal values in several patients, although T_1_ remained increased in two patients despite biochemical recovery, which may indicate progression to chronic kidney injury [81], as could be expected with AKI.

Several research groups have reported good repeatability of T_1_ data with less than 5.1% CV reported for both cortical and medullary T_1_ test–retest and inter-observer evaluations [46,48,81,120,137,143,144]. Such good repeatability is vital in drug development and suggests that T_1_ could potentially be used as a marker of kidney damage in drug development.

#### 7.4.2. DWI

Diffusion of water molecules within biological tissues depends on the intravascular, extracellular, or intracellular fluid compartments in which the molecules are located. Renal DWI probes the water molecules’ diffusion and interactions with tissue components and cell boundaries. Diffusivity can be altered by changes in the extracellular milieu due to fibrosis and associated collagen accumulation, oedema, cellular infiltration, or perfusion changes. The DWI-generated ADC can potentially separate patients with relatively “low level” fibrosis from “high level” fibrosis [81,139,140,141,145], making it a very promising biomarker for the assessment of fibrosis that itself is an important surrogate marker of DKD progression. ADC outperformed GFR in predicting fibrosis progression in kidney allografts patients who have undergone repeated biopsy and DWI assessments [146].

Renal ADC has been shown to correlate with interstitial fibrosis in several studies [147,148,149], although less is known about the exact underlying processes affecting the DWI signal [150]. The correlation between ADC and established biochemical biomarkers of kidney function and damage (i.e., GFR and UACR) in DKD patients has been confirmed in the univariate analysis of Makvandi et al. [48]. By adding mGFR as one variable, however, ADC was no longer significant in the bivariate prediction of UACR, suggesting that ADC could be linked to UACR via its effect on GFR. Interestingly, the ADC remained unchanged during 3 months of therapy in patients with renovascular disease and mild fibrosis and did not correlate with eGFR, serum creatinine, renal hypoxia, or inflammation, which could suggest that these factors are not captured in the ADC values [147]. In a sub-study of the COMBINE trial, baseline cortical ADC significantly correlated with the annual, patient-specific eGFR slope over 12 months (*p* = 0.08), although this was no longer significant once albuminuria was adjusted for, suggesting that there is either overlapping information between ADC and traditional risk factors for CKD progression, or albuminuria is a causal pathway to CKD progression [151]. 

Recently, PARENCHIMA published consensus-based guidance and recommendations to harmonize inter-site DWI MR protocols [54] and to facilitate translation of DWI in multicentre clinical trials [43]. Studies are also underway to confirm the link between renal DWI biomarkers and kidney fibrosis in DM subjects, such as the Biomarker Enterprise to Attack Diabetic Kidney Disease (BEAt-DKD) trial (NCT03716401) [152]. Another important factor for DWI use in clinical trials is based on its repeatability and reproducibility, which enables patient numbers to be reduced whilst still capturing changes due to treatment response. An intra-subject ADC CV of ≤ 7.2% has been reported in ADC test–retest repeatability analyses, which is less than the variability measured in UACR measurements [48]. Together, these studies support the efforts of ongoing multicentre initiatives to standardize methods and make DWI comparable between and deliverable at most imaging centres.

### 7.5. Magnetic Resonance Elastography

Magnetic resonance elastography (MRE) enables the study of tissue stiffness and is of particular interest for pathologies involving fibrosis like DKD. MRE has several similarities to other MR-based techniques used to study the diseased kidney while also being fundamentally different. Most of the techniques we have discussed rely solely on the imaging pulse sequence design to investigate an intrinsic tissue property of choice. With MRE, however, the MR scanner is used to monitor how externally applied mechanical waves propagate through the tissue of interest since wave velocity increases with increasing tissue stiffness [153].

The most common and clinically available implementation of MRE involves a pneumatically driven actuator vibrating at a specific frequency that is positioned over the anatomical area to be examined. An alternative and promising technique is called MR tomoelastography; this involves multiple actuators positioned around the area of interest that is reported to yield enhanced depth coverage compared to the single source approach [154]. To the best of our knowledge, MR tomoelastography is under development and not yet approved for clinical use. Notably, stiffness measurements based on the elastography principle are not limited to MRI but are also available with ultrasound-based techniques [155].

MRE offers a completely non-invasive examination and compared to renal biopsy, where only a limited number of points are sampled, a successful MRE examination provides a comprehensive overview of how the stiffness is distributed throughout the whole kidney. Thus, and considering that MRE stiffness is frequently used to monitor fibrotic development in liver diseases [156,157], MRE might also fulfil a similar role for the kidney and DKD. Two studies on kidney allografts have reported a positive correlation between fibrosis and stiffness [158,159], although, in two other studies on native kidneys with CKD (using MRE) [160] and DKD (using ultrasound elastography) [161], renal tissue stiffness actually decreased with progressing kidney disease and corresponding biopsy-determined increase in fibrosis. Chen et al. [161] speculated that the decrease in stiffness may have been due to hypoperfusion.

Healthy volunteer reproducibility studies on kidney stiffness with MRE have reported high concordance correlations between same day repeat scans (CV 3–10% depending on MRE technique and cortex or medulla) [162] and a mean difference of 6% for intra-subject measurements performed 4–5 weeks apart [163]. A corresponding study for the MR tomoelastography technique reported an intra-class correlation for reproducibility coefficient of 0.78 for the whole parenchyma [164]. In a study on patients with IgAN, renal stiffness measured by MR tomoelastography was compared to ADC and BOLD, and renal stiffness demonstrated a greater sensitivity to IgAN than ADC (AUC 0.9 vs. 0.8), whereas BOLD showed no effect [165].

While a change in renal impairment in DKD can be detected by a change in renal stiffness, and MRE stiffness measurements have demonstrated reproducibility, the parameter should probably still be used with caution in DKD clinical studies since fibrosis and the other changes that progress with DKD may mask or suppress each other’s impact on renal stiffness.

### 7.6. Fatty Kidney

Fatty kidney comprises perirenal, sinus, and renal parenchyma fat [166,167,168,169]. Perirenal and sinus fat volumes can be measured with a variety of high-resolution MRI techniques to visualize the renal facia that separate perirenal fat from the pararenal depot. Renal sinus fat is an extension of the perirenal fat in close contact with the renal pelvis and calyces. Renal parenchyma fat comprises the lipid droplets inside the renal parenchymal cells and can be measured with either the proton density fat fraction (PDFF) imaging technique [170] or a localized spectroscopy technique [171]. PDFF produces a map showing the proportion of fat in each voxel of the image. The localized spectroscopy technique isolates the proton MR spectroscopy signal from a selected region of the body and can be used to quantitate signals from the lipid methyl, methylene, and protons associated with double bonds.

Perirenal fat is of interest as a target for drug development, as it may contribute to the pathogenesis of hypertension, obesity, and chronic renal diseases [172]. Excessive perirenal and sinus fat may induce mechanical compression of the renal parenchyma, vasculatures, nerve fibres, and the collecting system and contribute to hyperfiltration and hypertension [173]. A larger sinus fat volume has been found in T2DM patients compared to healthy controls and is associated with HbA1c, visceral adipose tissue (VAT), and blood lipids [174], yet treatment with intensive glucose-lowering therapy had no effect on sinus fat volume despite the significant decrease in HbA1c, VAT, triglyceride, and cholesterol (Table 1) [75]. The contributions of these three types of renal fat to CKD and DKD have recently been reviewed [175]. 

Perirenal and sinus fat can be measured with CT [176], ultrasound [177], and MRI [178]. Ultrasound can measure the thickness of perirenal fat on the lateral aspect of the abdomen, although this approach may also include pararenal fat [175]. Some perirenal fat studies have used CT to measure perirenal fat thickness or volume [175]; however, perirenal fat is not homogeneous around the kidney, and there is no consensus on where to measure perirenal fat thickness [175,176]. Renal sinus fat has also been measured as an area in a single kidney slice or as volume from multiple slices. Given the small size and irregular shape of renal sinus fat, volumetric measurement of renal sinus fat is recommended, as well as using MRI rather than CT to minimize the risks from ionizing radiation from the CT scan [175].

MRI is the only non-invasive technique to measure renal parenchyma fat. Localized spectroscopy is a specialized technique that may be difficult to run in a multicentre clinical trial, whereas PDFF is a common endpoint in liver studies to quantify hepatic fat content. The challenge in the kidney is that renal parenchymal fat exists at much lower levels than in the liver (commonly 1–3%), and, in people with DM, the kidney is commonly surrounded by a thick layer of perirenal, pararenal, and other visceral fat depots. Thus, it is important to design a PDFF protocol that minimizes artefacts originating outside of the kidney that could contaminate the renal PDFF signal. PDFF methods have shown significantly higher levels of renal parenchymal fat in obese individuals [179], T2DM patients [180], and T2DM patients with microalbuminuria [181] compared to relevant control groups. Intervention studies in T2DM patients have shown significant reductions in renal parenchyma fat after either 26 weeks of treatment with liraglutide [171] or at least 3 months of treatment with empagliflozin [182].

### 7.7. Multiparametric Imaging

In many diseases, univariate measurements do not entirely describe the disease or efficacy of treatment effects. mpMRI incorporates multiple anatomical and functional quantitative imaging biomarkers allowing more comprehensive tissue characterization than any single measure [183]. Combining mpMRI endpoints into a single determination of longitudinal change has mostly been limited to the development of composite endpoints that use logic operators, such as “AND” and “OR”, to determine a binary outcome based on a priori-defined thresholds applied to the components of the composite endpoint. One example is PI-RADS V2.1 for assessment of prostate cancer [184]. An alternative approach is to represent multiple MRI biomarkers as a single multidimensional vector to represent the disease state more completely [183]. This mpMRI descriptor may be thought of as providing an mpMRI signature of the disease state and may one day be capable of differentiating between DKD and other forms of CKD (underlying IgA nephropathy for example). Future studies will show whether mpMRI signatures can be defined for DKD and other kidney diseases and if changes in the mpMRI vector can be used to track clinical interventions with novel therapeutics.

### 7.8. Multi-Organ Imaging

As many comorbidities can drive multi-organ dysfunction that may, in turn, lead to CKD, a “whole-body” investigative approach may be appropriate for CKD and its treatment. The impact of pharmacological treatment is usually assessed with circulating biomarkers, although it can be difficult to link such biomarkers to the individual organ of interest due to their whole body/systemic nature. The advantage of MRI is that it generates spatial information. Thus, MRI has the capability to investigate not only the individual organ affected, such as the kidney, but other organs that might be affected and provide a whole-body, systemic overview (Figure 3).

This does not necessarily mean that the current regulatory landscape would allow a single, pivotal clinical trial to be used for registration of protection of multiple end-organs, including the kidney. The use of whole-body MRI is complicated by clinical trial inclusion/exclusion criteria specific to the individual disease phenotype. Patient characteristics also may differ between populations (i.e., a CKD population is usually much older than a typical population with non-alcoholic steatohepatitis), which further hinders inclusion of two or more diseases in the same population. Thus, MRI of multiple organ endpoints is primarily used in early phase clinical trials (e.g., phases 1B and 2A) to provide information on potential clinical and sub-clinical improvements on end-organ status induced by novel therapies. The increased cost and complexity of including MRI in a clinical trial means that MRI should ideally be used when circulating biomarkers are not sufficient to answer the biological questions defined in the clinical study.

## 8. Challenges of Multi-Site Studies

The renal MRI techniques described in this review are not currently used for diagnostic MRI of the kidney, and the vast majority of the studies discussed in this review are academic, single-centre studies. However, these methods are now starting to be applied in multicentre studies [152,185,186]. Clinical trials typically involve several imaging sites and potentially hundreds of sites. Most commonly, a central imaging facility that may be part of the sponsoring company or a partner contract research organization (CRO) is responsible for coordinating image acquisition and processing. The importance of standardization has been recognized both by the regulators and industry to ensure the quality of output. Regulators have issued guidance on the standards that should be applied when designing imaging clinical trials [62], and consensus statements on best practices in clinical trials [43,187] offer additional guidance.

The imaging charter [62] is a formal document detailing the roles and responsibilities of the various parties involved in a clinical trial and includes imaging acquisition, display, archiving, and interpretation process standards. It is developed together with the sponsor. Other important imaging-specific documents include the “Imaging Manual” and the “Corelab Manual”, which describe how images are acquired at the imaging sites and subsequently analysed at the Core Imaging Facility (CIF). An important consideration when developing the documentation is to decide the best balance between fidelity and feasibility. Radiology departments have MRI scanners from different manufacturers used at different magnetic field strengths, so restricting the study to similar MRI scanners across all imaging sites may not be feasible. In this case, the CIF specifies a harmonized imaging protocol to control potential sources of imaging bias and variability. The main activities of the CIF are to establish the sites where the data are collected, to verify that the data conform to the required standard (site qualification), and to provide ongoing support and quality assurance throughout the trial (Figure 4).

The task of imaging specialists, either within the sponsoring company or outsourced through a CRO, is to implement MRI in a clinical trial. The overwhelming majority—if not all—of MRI trials are performed in collaborating clinical radiology departments. The need for reproducibility within a site and consistency between sites are contradictory to the needs of diagnostic imaging, which is concerned with producing the best quality image to visualize a specific pathology in a given individual. In clinical trials, however, the MRI protocol is locked for the full duration of the study, and only pre-specified changes within a limited range are allowed. 

Imaging site personnel must be trained to implement a standardized imaging protocol that is applied by all clinical sites participating in the trial. After training, each site must submit test data for review by the CIF. Once a data set has been approved, the clinical site receives a formal authorization letter to start scanning patients for the clinical trial. Quality control (QC) of incoming images is extremely important; if any image fails a QC evaluation, a repeat examination must be requested, detailing the shortcomings of the images.

## 9. Conclusions

The rapid evolution of MRI techniques over the past decade has revealed new opportunities to non-invasively assess and quantify functional, structural, and pathophysiological changes in DKD and offer a more personalized approach to its management. Several pioneering studies have demonstrated that advanced mpMRI tools provide more comprehensive information across individual kidney compartments on microstructure (including kidney fibrosis and inflammation), macrostructure (kidney volume), oxygenation, and haemodynamic measurements of RBF and perfusion. This may improve our understanding of different phenotypes and progression of DKD and, coupled with its higher reproducibility and potential to lower the overall cost of clinical trials, enable a more comprehensive analysis of efficacy and modes of action of potential therapeutic interventions for DKD. MRI biomarkers are likely to play a strong role as biomarkers in future clinical drug trials in DKD patients and improve their managements and prognoses.

The imaging methods discussed in this review all offer unique insights into kidney physiology and pathology. However, one of the constraints for imaging studies is the length of time that subjects spend in the scanner. This should be as short as possible, and, therefore, it is necessary to select imaging methods appropriate to the target of interest in the clinical trial. One combination that has proved useful in a range of studies and that can be performed in under 40 min is T1 mapping to provide insight into kidney microstructures and cortical and parenchymal volumes, PC-MRI to provide haemodynamics, BOLD MRI as a marker of oxygenation, and DWI as a marker of fibrosis. If the target of interest is metabolic, then PDFF to measure kidney parenchymal fat can be acquired in an extra 5 min. Techniques such as ASL and DCE-MRI can be used in expert single-centre studies if the extra time for the acquisition can be justified. 

Nevertheless, several challenges remain to be addressed before these techniques become commonplace in clinical practice for the diagnosis of DKD. Rigorous technical and clinical validations are needed, although the major limitation is likely to be the provision of evidence of clinical utility to show that MRI provides important information about the disease or the intervention that is not available by other means. Prognostic biomarkers are needed by clinicians to identify the likelihood of a clinical event, disease recurrence or progression in patients, and to identify fast progressors for inclusion in clinical trials. In addition, cost-effectiveness must be demonstrated. These challenges need to be balanced against the huge potential benefit the DKD population is likely to experience from the advances in these imaging techniques because of the complex and heterogeneous nature of the disease.

Advances in kidney imaging are continuing to be driven by consortia, such as the EU IMI project BEAt-DKD [188], the UK Renal Imaging Network [189], the Kidney Precision Medicine Project [190], and RENALMRI.org (renalmri.org), as well as academics and pharmaceutical companies interested in the validation of novel biomarkers of kidney disease. One of the aims of these consortia is to continue the work started by PARENCHIMA on the harmonization of kidney MRI acquisition and analysis methods. Harmonization will drive the implementation of improved methods by scanner manufacturers, the development of image analysis techniques, and acceptance by clinicians and regulators. 

## Figures and Tables

**Figure 1 jcm-12-04625-f001:**
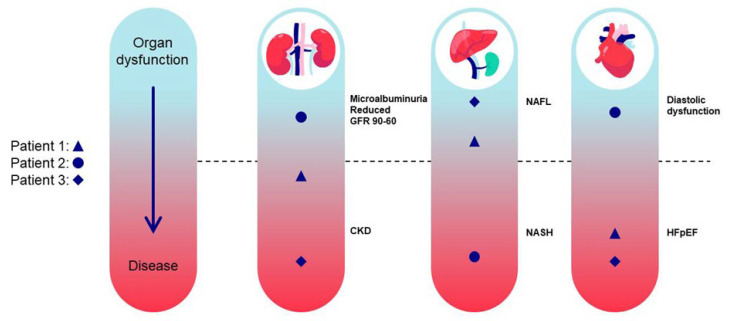
The Course to End-Organ Failure with Overlapping End-Organ Dysfunction. Individual differences in overlapping disease states exemplified in three different patients and their progression from organ dysfunction to end-stage organ disease, for example, microalbuminuria and reduced GFR to CKD. CKD = chronic kidney disease; GFR = glomerular filtration rate; HFpEF = heart failure with preserved ejection fraction; NAFL = non-alcoholic fatty liver; NASH = non-alcoholic steatohepatitis.

**Figure 3 jcm-12-04625-f003:**
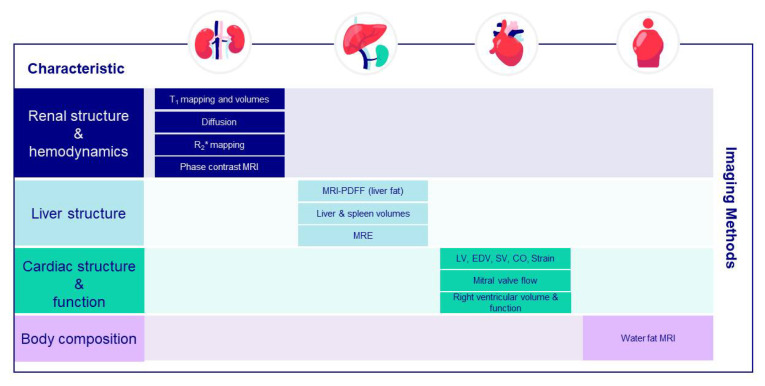
An Integrated Imaging Protocol of End-Organ Damage and Disease. An integrated imaging protocol for the kidneys, liver, heart, and body composition that can be performed using multi-organ imaging methodology in under 45 min. CO = cardiac output; EDV = end-diastolic volume, LV = left ventricle; MRE = magnetic resonance elastography; MRI = magnetic resonance imaging; MRI-PDFF = magnetic resonance imaging-derived proton density fat fraction; SV = stroke volume.

**Figure 4 jcm-12-04625-f004:**
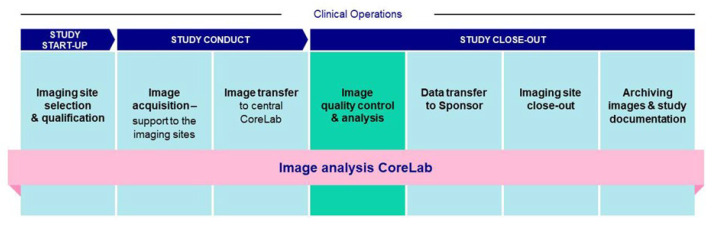
The Main Activities Performed by the Core Imaging Facility. Image analysis flow through the Core Imaging Facility CoreLab during a clinical study/trial using MRI.

**Table 1 jcm-12-04625-t001:** Selected Intervention Studies using Renal MRI Endpoints.

Reference	Intervention	Subjects	N	MR Intervention	Comments
Prasad et al., 1996 [66]	Furosemide; acetazolamide; water loading	HV	7	BOLD	Furosemide and water diuresis decreased medullary R_2_*
Manotham et al., 2012 [67]	Olmesartan	DKD; CKD; HV	19	BOLD	Olmesartan decreased R_2_* in both CKD groups, but not in HV after 60 min
Prasad et al., 2015 [68]	Furosemide	CKD	59	BOLD, DWI, Volume	CKD patients had lower renal volume, higher cortical R_2_*, and lower response to furosemide on medullary R_2_*
Vakilzadeh et al., 2015 [69]	Aliskiren; HCTZ	HT	20	BOLD	Aliskiren and HCTZ patients with a fall in systolic blood pressure > 10 mmHg decreased cortical R_2_* levels
Vink 2015 [70]	Captopril	HT	75	BOLD	The blood pressure-lowering effect of captopril was positively related to cortical and medullary R_2_*
Vakilzadeh et al., 2019 [71]	Glucose	Overweight	19	BOLD	Acute hyperglycemia decreased renal R_2_*
Khatir et al., 2019 [72]	RAS inhibitors/CCB; metoprolol	CKD	75	PC-MRI, BOLD	Vasodilation treatment reduced intrarenal vascular resistance but did not affect R_2_*
Laursen et al., 2021 [73]	Dapagliflozin; placebo	T1DM + albuminuria	15	BOLD, ASL, PC-MRI	A single dose of dapagliflozin decreased cortical R_2_* without changes in renal perfusion or blood flow
Lee et al., 2022 [74]	Empagliflozin; placebo	Heart failure	105	ASL, T_1_, apparent extracellular volume, post-contrast T_1_, TKV	Empagliflozin reduced perfusion (ASL) and kidney extracellular volume. No between-group differences in kidney T_1_, TKV, or eGFR
Lin et al., 2023 [75]	Liraglutide; placebo	T2DM	96	Renal volume, sinus fat volume	Renal enlargement in T2DM can be reversed by liraglutide treatment
Gullaksen et al., 2023 [76]	Semaglutide; empagliflozin; combination; or placebo	T2DM + CVD risk	80	BOLD, ASL	Empagliflozin increased medullary R_2_*; semaglutide decreased perfusion in cortex and medulla

ASL = arterial spin labelling; BOLD = blood oxygenation level-dependent; CCB = calcium channel blockers; CKD = chronic kidney disease; CVD = cardiovascular disease; DKD = diabetic kidney disease; DWI = diffusion-weighted imaging; eGFR = calculated glomerular filtration rate; HCTZ = hydrochlorothiazide; HT = hypertension; HV = healthy volunteers; MRI = magnetic resonance imaging; PC-MRI = phase contrast imaging; R_2_* = BOLD MRI relaxation rate; RAS = renin-angiotensin system; T_1_ = relaxation time/relaxation rate R_1_; T1DM = type 1 diabetes mellitus; T2DM = type 2 diabetes mellitus; TKV = total kidney volume.

## Data Availability

Not applicable.

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
