# Peer review of "Magnetic Resonance Imaging in Clinical Trials of Diabetic Kidney Disease"

_jcm, 2023, doi:10.3390/jcm12144625_

Round 1

Reviewer 1 Report

This is a timely review article.  There is growing interest in multiparametric MRI in the evaluation of kidneys both in health and disease.  While much of the interest was spurred by academic interests, with the growing number of novel interventions that have become available in the last few years, there is now a need for tools to evaluate the efficacy of these drugs in clinical trials.  This is especially true given that the mechanism behind the renal outcomes associated with SGLT2i and GLP agents is not yet clear. 

Application to clinical trials brings in a whole new set requirements.  The ability to harmonize acquisitions across multiple vendor platforms, the level of reproducibility, have reproducible analysis techniques etc..  The authors have done a good job summarizing the current state-of-the-art for several of the potential techniques and their suitability for use in clinical trials.  They can consider to strengthen it further by addressing the following.

Specific Comments:

1.      It would be useful for the authors to summarize what they believe are viable set of methods that can be used in clinical trials now

2.      Propose a short list of challenges for the community to be aware and hopefully contribute towards.  Is prediction markers for progression a target to strive for?

3.      The authors should consider including a discussion on harmonization of analysis methods for MP-MRI.  I believe ultimately it may involve AI to automatically segment the kidney parenchyma.  However, there needs to be ground work to train AI.  Further, efficient registration techniques are necessary to use the segmented ROIs across multiparametric data. 

4.      Is there a potential cost benefit of imaging based markers compared to conventional markers in terms of requiring smaller n?

Author Response

Many thanks for the very helpful comments. In view of the length of the review we have tried to keep additional text to a minimum. 

  1. It would be useful for the authors to summarize what they believe are viable set of methods that can be used in clinical trials now
    Paragraph added to conclusions (lines 873-883).
  2. Propose a short list of challenges for the community to be aware and hopefully contribute towards.  Is prediction markers for progression a target to strive for?
    Sentence added to conclusions (lines 888-890).
  3. The authors should consider including a discussion on harmonization of analysis methods for MP-MRI.  I believe ultimately it may involve AI to automatically segment the kidney parenchyma.  However, there needs to be ground work to train AI.  Further, efficient registration techniques are necessary to use the segmented ROIs across multiparametric data. 
    Two sentences added to conclusions (lines 899-902).
  4. Is there a potential cost benefit of imaging based markers compared to conventional markers in terms of requiring smaller n?
    There is an existing discussion on lines 310-320. Is this sufficient or would the reviewer like us to expand this section?

Reviewer 2 Report

General remarks:

This is a very comprehensive overview of what renal MRI has to offer to patients with CKD, in particular DKD. In fact, it is one of the most comprehensive reviews I have read on the subject. The authors describe very well the different existing MR techniques, and summarize the landmark studies performed in the field. They also clearly indicate what role renal MRI can play in clinical trials. There is not much to improve, except that the article could be shortened. 

I would advise two changes:

-the article is very long, especially the introduction, and it would be good to shorten it a bit , in order not to lose the readers halfway, as the end of the review is very strong. For the introduction, to my opinion, line 91-155 can be suppressed, and probably other parts as well. They could refer to review articles on DKD that describe in more detail the pathophysiology of DKD. Other parts can also be shortened.

-instead, further in the article, it would be nice if the authors could express their thoughts on the possibility that mpMRI may one day be capable of differentiating between DKD and other forms of CKD (underlying IgA nephropathy for example). One paragraph on this subject would be very nice. Authors should be prudent here, as not many data support this possibility thus far, but it is definitely a subject that merits more research.

Minor remarks:

Line 66-67: it is true that tubule-interstitial fibrosis is a better predictor of renal function decline than glomerular pathology, but this does not hold for all forms of CKD (rapidly progressive GN..), and other forms are found in 10-30% of biopsies performed in patients with diabetes. SO maybe specify that this concerns basically DKD.

Line 77-78 DKD patients are still staged prognostically…

Maybe better say: DKD patients are still mostly staged prognostically based on the basis of ..

Let’s not forget KFRE scores, that also include calcium, phosphate and albumin.

-line 91-106 can be deleted to my opinion, including the figure of CKD stages that is well known by all.

-paragraph 4: I would add “until recently” to line 164.

-PC-MRI: underline that this technique can measure RBF of each individual kidney ) stenotic vs non stenotic kidney for ex).

-BOLD-MRI: a little word on relative blood volume of voxels, on how it impacts the interpretation of R2* and recent solutions (Prasad, KI reports 2023) would be welcome, if there is enough space.

-I like paragraphs 7.8 and 8, as we don’t read much about this in other reviews.

-Figure 3: not all the discussed techniques are integrated in this figure (I didn’t see DCE-MRI, MRE and PDFF in it, for ex).

Author Response

Many thanks for the insightful comments. I have attempted to respond to the reviewers comments below (please see below).

I would advise two changes:

-the article is very long, especially the introduction, and it would be good to shorten it a bit , in order not to lose the readers halfway, as the end of the review is very strong. For the introduction, to my opinion, line 91-155 can be suppressed, and probably other parts as well. They could refer to review articles on DKD that describe in more detail the pathophysiology of DKD. Other parts can also be shortened.
Many thanks for the suggestion. We have deleted lines 94-110 plus Figure 1, and refer to the relevant papers instead. We would prefer to retain lines 111-155 as we refer to the DKD phenotype discussion later in the review and imaging experts will not be aware of the various phenotypes.

-instead, further in the article, it would be nice if the authors could express their thoughts on the possibility that mpMRI may one day be capable of differentiating between DKD and other forms of CKD (underlying IgA nephropathy for example). One paragraph on this subject would be very nice. Authors should be prudent here, as not many data support this possibility thus far, but it is definitely a subject that merits more research.
Many thanks for this helpful suggestion. We have added text to section 7.7 to exemplify this point (lines 555-561).

Minor remarks:

Line 66-67: it is true that tubule-interstitial fibrosis is a better predictor of renal function decline than glomerular pathology, but this does not hold for all forms of CKD (rapidly progressive GN..), and other forms are found in 10-30% of biopsies performed in patients with diabetes. SO maybe specify that this concerns basically DKD.

Changed.

Line 77-78 DKD patients are still staged prognostically…

Maybe better say: DKD patients are still mostly staged prognostically based on the basis of ..
Thank you for the improvement. Changed.

Let’s not forget KFRE scores, that also include calcium, phosphate and albumin.

-line 91-106 can be deleted to my opinion, including the figure of CKD stages that is well known by all.

Thanks. Deleted.

-paragraph 4: I would add “until recently” to line 164.
Thank you. Changed.

-PC-MRI: underline that this technique can measure RBF of each individual kidney) stenotic vs non stenotic kidney for ex).
Text added on lines 434-437.

-BOLD-MRI: a little word on relative blood volume of voxels, on how it impacts the interpretation of R2* and recent solutions (Prasad, KI reports 2023) would be welcome, if there is enough space.
Two sentences added to section 7.3.

-I like paragraphs 7.8 and 8, as we don’t read much about this in other reviews.

Thank you.

-Figure 3: not all the discussed techniques are integrated in this figure (I didn’t see DCE-MRI, MRE and PDFF in it, for ex).

This is now Figure 2: Changed to “Recommended MRI techniques”. ASL removed and (optional) kidney PDFF added.